# A Novel System for Spinal Muscular Atrophy Screening in Newborns: Japanese Pilot Study

**DOI:** 10.3390/ijns5040041

**Published:** 2019-11-12

**Authors:** Masakazu Shinohara, Emma Tabe Eko Niba, Yogik Onky Silvana Wijaya, Izumi Takayama, Chisako Mitsuishi, Sakae Kumasaka, Yoichi Kondo, Akihiro Takatera, Isamu Hokuto, Ichiro Morioka, Kazutaka Ogiwara, Kimimasa Tobita, Atsuko Takeuchi, Hisahide Nishio

**Affiliations:** 1Department of Community Medicine and Social Healthcare Science, Division of Epidemiology, Kobe University Graduate School of Medicine, 7-5-1 Kusunoki-cho, Chuo-ku, Kobe 650-0017, Japan; mashino@med.kobe-u.ac.jp (M.S.); niba@med.kobe-u.ac.jp (E.T.E.N.); yogik.onky@gmail.com (Y.O.S.W.); itakayam@med.kobe-u.ac.jp (I.T.); 2Japanese Red Cross Katsushika Maternity Hospital, 5-11-12 Tateishi, Katsushika-ku, Tokyo 124-0012, Japan; mitsuishi@katsushika.jrc.or.jp (C.M.); kumasaka@nms.ac.jp (S.K.); 3Matsuyama Red Cross Hospital, 1 Bunkyo-cho, Matsuyama 790-8524, Japan; ykondoh@matsuyama.jrc.or.jp; 4Chibune General Hospital, 3-2-39 Fukumachi, Nishiyodogawa-ku, Osaka 555-0034, Japan; a-takatera@kakohp.jp; 5Department of Pediatrics, St. Marianna University School of Medicine, 2-16-1 Sugao, Kawasaki 216-8511, Japan; isamuhokuto@gmail.com; 6Department of Pediatrics and Child Health, Nihon University School of Medicine, 30-1 Oyaguchi kamicho, Itabashi-ku, Tokyo 173-8610, Japan; morioka.ichiro@nihon-u.ac.jp; 7Biogen Japan Ltd., 1-4-1 Nihonbashi, Chuo-ku, Tokyo 103-0027, Japan; cntcr303@ybb.ne.jp (K.O.); ktobita1103@gmail.com (K.T.); 8Kobe Pharmaceutical University, 4-19-1, Motoyamakitamachi, Higashinada-ku, Kobe 658-8558, Japan; takeuchi@kobepharma-u.ac.jp; 9Department of Occupational Therapy, Faculty of Rehabilitation, Kobe Gakuin University, 518 Arise, Ikawadani-cho, Nishi-ku, Kobe 651-2180, Japan

**Keywords:** mCOP-PCR, *SMN1*, *SMN2*, spinal muscular atrophy

## Abstract

Spinal muscular atrophy (SMA) is a neuromuscular disorder caused by *SMN1* gene deletion/mutation. The drug nusinersen modifies *SMN2* mRNA splicing, increasing the production of the full-length SMN protein. Recent studies have demonstrated the beneficial effects of nusinersen in patients with SMA, particularly when treated in early infancy. Because nusinersen treatment can alter disease trajectory, there is a strong rationale for newborn screening. In the current study, we validated the accuracy of a new system for detecting *SMN1* deletion (Japanese patent application No. 2017-196967, PCT/JP2018/37732) using dried blood spots (DBS) from 50 patients with genetically confirmed SMA and 50 controls. Our system consists of two steps: (1) targeted pre-amplification of *SMN* genes by direct polymerase chain reaction (PCR) and (2) detection of *SMN1* deletion by real-time modified competitive oligonucleotide priming-PCR (mCOP-PCR) using the pre-amplified products. Compared with PCR analysis results of freshly collected blood samples, our system exhibited a sensitivity of 1.00 (95% confidence interval [CI] 0.96–1.00) and a specificity of 1.00 (95% CI 0.96–1.00). We also conducted a prospective SMA screening study using DBS from 4157 Japanese newborns. All DBS tested negative, and there were no screening failures. Our results indicate that the new system can be reliably used in SMA newborn screening.

## 1. Introduction

Spinal muscular atrophy (SMA) is an autosomal recessive neuromuscular disorder characterized by degeneration of motor neurons in the spinal cord, which results in progressive muscle atrophy and weakness [1]. With a reported incidence of approximately 1 in 6000 to 10,000 live births [2], SMA is the second most common fatal autosomal recessive disorder after cystic fibrosis [3]. SMA is considered to be the leading genetic cause of infant death [1]. Patients with SMA type 1, the severe phenotype, develop symptoms in the first 6 months after birth, never achieve the motor milestone of sitting independently, and have a life expectancy of less than 2 years without respiratory support [1,4,5].

The survival motor neuron (*SMN*) genes located on chromosome 5q13, *SMN1* and *SMN2*, were identified as the SMA-related genes in 1995 [6]. *SMN1* and *SMN2* are paralogs and are almost identical except for five nucleotides [6]. *SMN1* expresses the full-length transcript and results in the production of the functional, full-length SMN protein [6]. The functional, full-length SMN protein plays a critical role in RNA metabolism and other cellular functions [7]. In contrast, *SMN2* expresses two kinds of transcripts, the major one is an exon 7-skipped transcript (Δ7-transcript) due to a C-to-T change in exon 7, producing a non-functional, truncated SMN protein. The minor transcript is a full-length transcript encoding the same protein derived from *SMN1*. The presence of *SMN2* does not fully compensate for the loss of *SMN1* because *SMN2* can only produce a limited amount of the full-length SMN protein [1,7].

*SMN1* is absent (or homozygously deleted) in approximately 95% of patients with SMA and deleteriously mutated in the remaining patients [6]. On the other hand, a higher copy number of *SMN2* is associated with a milder phenotype of SMA [8]. *SMN1* is proven to be a disease-causing gene, while *SMN2* is now considered to be a disease-modifying gene [6,8]. Therefore, the absence of both genes causes embryonic lethality in mice [9,10]. In fact, all patients with homozygous deletion of *SMN1* retain at least one copy of the *SMN2* gene [1].

SMA was considered to be an incurable disease. Nevertheless, intrathecal administration of an antisense-oligonucleotide drug, nusinersen, has been associated with encouraging clinical efficacy in SMA patients [11,12]. The drug modifies *SMN2* splicing and increases the production of the functional, full-length SMN protein [11,13]. Nusinersen has been approved by regulatory agencies in multiple countries [the United States Food and Drug Administration (2016), the European Medicines Agency (2017), the Ministry of Health, Labor and Welfare of Japan (2017), the Ministry of Food and Drug Safety of the Republic of Korea (2018), and the China National Medical Products Association (2019)].

Treatment of SMA patients with nusinersen also appears to result in a better clinical outcome when it is initiated in early infancy [11,12]. Early diagnosis and initiation of treatment, ideally before apparent symptoms develop, may be important for the optimal response to nusinersen [11]. However, without newborn screening for SMA, treatment cannot be initiated until a significant number of motor neurons have been lost [14].

Ideally, implementation of newborn screening programs for SMA would allow pre-symptomatic diagnosis of the disease in many cases and the early initiation of treatment with potential for maximal therapeutic benefit [15]. In the USA, SMA is now included in the recommended uniform screening panel (RUSP), and newborn screening has been implemented in screening programs in a number of states [16,17].

We have developed a rapid, accurate, and high-throughput system for detecting *SMN1* deletion using a real-time modified competitive oligonucleotide priming-polymerase chain reaction (mCOP-PCR) technique combined with targeted pre-amplification of the *SMN* genes from dried blood spots (DBS) [18]. Here, we describe the results of a pilot study of newborn screening for SMA to validate our system and to genotype all DBS collected from newborns in Japan.

## 2. Materials and Methods

### 2.1. Objectives and Ethics

The primary objective of this pilot study was to validate the accuracy of our newly developed *SMN1*-deletion detection system using DBS from individuals with and without genetically confirmed SMA. An additional objective was to investigate the accuracy of the *SMN1*-deletion detection system for prospective newborn SMA screening using DBS. The study was approved by the institutional review boards at all participating hospitals, as well as the Ethics Committee of the Kobe University Graduate School of Medicine (reference 170165, approved on 27November, 2017), and was conducted in accordance with the World Medical Association Declaration of Helsinki.

### 2.2. SMA Patients and Non-SMA Controls

The *SMN* genotype of the patients and controls had been previously analyzed by PCR–restriction fragment length polymorphism (PCR–RFLP) using extracted DNA from freshly collected blood. PCR–RFLP was carried out according to the method of van der Steege and colleagues [19]. DBS from the same patients and controls were stored in the sample library at the Division of Epidemiology, Kobe University Graduate School of Medicine, Japan. DBS from 100 individuals (50 SMA patients and 50 non-SMA controls) were analyzed using the new *SMN1*-deletion detection system. Written informed consent for the use of all DNA samples was obtained from the patients/controls and/or their parents.

### 2.3. Newborn Infants

Infants born at 49 hospitals across Japan were eligible to participate in the pilot screening study. Written informed consent for study participation was obtained from parents/guardians of newborn infants. Information about the study was provided, and it was explained that participation was voluntary.

### 2.4. SMN1-Deletion Detection System

As illustrated in Figure 1, our system consisted of two steps, namely, (1) targeted pre-amplification of the *SMN* genes: the target sequences of the *SMN* genes from DBS were pre-amplified by conventional PCR, and (2) gene-specific amplification of *SMN1* and *SMN2* exon 7: *SMN1*-deletion was detected by mCOP-PCR with the pre-amplified products (Japanese patent application No. 2017-196967, PCT/JP2018/37732). In the previous version of our system, genomic DNA was extracted from DBS before targeted pre-amplification [18]. However, the new version used in the present study did not include the DNA extraction step. Instead, a punched circle from each DBS was placed directly into the reaction tube of the conventional PCR (direct PCR) [20].

#### 2.4.1. Targeted Pre-Amplification of the SMN Genes

Targeted pre-amplification of the sequence containing *SMN1*/*SMN2* exon 7 was performed with conventional PCR using the GeneAmp^®^ PCR System 9700 (Applied Biosytems, Foster City, CA, USA). A punched circle 2 mm in diameter (equivalent to ~15 µL of whole blood) from each DBS was added to the reaction mixture with DNA polymerase KOD FX Neo (TOYOBO, Osaka, Japan). The following primers were used to amplify the target sequence containing *SMN1*/*SMN2* exon 7: R111 (5′-AGA CTA TCA ACT TAA TTT CTG ATC A-3′) and 541C770 (5′-TAA GGA ATG TGA GCA CCT TCC TTC-3′) [6]. The PCR conditions for the 50 µL reaction mixture were: (1) initial denaturation at 94 °C for 7 min; (2) 40 cycles of denaturation at 94 °C for 1 min, annealing at 56 °C for 1 min, and extension at 72 °C for 1 min; (3) additional extension at 72 °C for 7 min; and (4) hold at 10 °C. The PCR product (i.e., the pre-amplified *SMN* gene product) was then subjected to gel electrophoresis and visualized using Midori-Green staining (NIPPON Genetics, Tokyo, Japan).

#### 2.4.2. Gene-Specific Amplification of SMN1 Exon 7

Real-time mCOP-PCR *SMN1* and *SMN2* exon 7 amplification was performed using the LightCycler^®^ 96 system (Roche Applied Science, Mannheim, Germany). An aliquot of pre-amplified PCR product was added to the reaction mixture with DNA polymerase KOD FX Neo (TOYOBO) and EvaGreen^®^ Dye (Biotium, Hayward, CA, USA). The primer set for *SMN1*-specific amplification consisted of R111 and SMN1-COP (5′-TGT CTG AAA CC-3′) [18,21]. The PCR conditions for the reaction mixture of 50 µL were: (1) initial denaturation at 94 °C for 7 min; (2) 20 cycles for *SMN1* denaturation at 94 °C for 1 min, annealing at 37 °C for 1 min, and extension at 72 °C for 1 min; and (3) melting analysis. Fluorescence signals were detected at the end of each extension procedure.

### 2.5. Newborn Screening Study Design

After obtaining written informed consent for SMA screening, approximately 100 µL of blood was collected from newborn infants and spotted onto filter paper (FTA^®^ Elute Cards, GE Healthcare, Boston, MA, USA) during standard blood sampling for tandem mass spectrometry newborn screening. DBS sample collection began at the end of January 2018 and ended at the end of April 2019.

The flow of DBS samples and data collection during the study is illustrated in Figure 2. Filter papers with DBS were transferred to Kobe University within 7 days after blood sampling and stored in the dark at room temperature (20–25 °C) until use. PCR experiments and the final data analysis were completed within 10 ± 4 days after the blood sampling. Any positive screening result (i.e., when *SMN1* deletion was detected) was verified by PCR–RFLP using the method of van der Steege et al. [19], and the primary physician of the infant was informed of the result so that the patient could be further examined, definitively diagnosed and, if necessary, treatment could be initiated. Any surplus specimen would be stored at the Division of Epidemiology, Kobe University Graduate School of Medicine, for up to 5 years.

### 2.6. Follow-Up Study of the Infants Screened for SMA

To determine the SMA status of the screened infants at 6 and 10 months after DBS collection, a survey was conducted among the physicians participating in the study, with a questionnaire including the question “Are there any patients diagnosed with SMA or SMA-like disease among the infants screened for SMA in this study?”.

### 2.7. Statistical Analysis

For validation of the screening system, 100 DNA samples (from 50 SMA patients and 50 controls) from the sample library of the Division of Epidemiology, Kobe University Graduate School of Medicine, were analyzed. The sensitivity and specificity of the screening system for the detection of *SMN1* exon 7 deletion were calculated using the original results of the PCR–RFLP analysis as the reference. The exact method was used to calculate 95% confidence intervals (CIs) for sensitivity and specificity [22,23]. Statistical analysis was performed using Epi Info (Centers for Disease Control, Atlanta, GA, USA).

## 3. Results

### 3.1. Validation Study

*SMN1* exon 7 and *SMN2* exon 7 differ in only one nucleotide at position 6, i.e., C in *SMN1* exon 7 and T in *SMN2*. For the detection of *SMN1* deletion, gene-specific amplification is essential. Figure 3 shows amplification of *SMN1* exon 7 by real-time mCOP-PCR with SMN1-COP primer. DBS samples with the *SMN1*(+) genotype showed marked amplification by real-time mCOP-PCR with SMN1-COP primer, whereas samples with the *SMN1*(–) genotype showed no amplification. These results indicated that our system was able to specifically amplify *SMN1* exon 7.

A total of 100 DBS samples from 100 individuals with and without genetically confirmed SMA were analyzed by real-time mCOP-PCR with SMN1-COP primer. They were from 50 SMA patients with *SMN1*(–) genotype and from 50 controls with *SMN1*(+) genotype. Figure 4 shows that the quantitation cycle values (Cq values) of 50 SMA patients with *SMN1*(–) were markedly higher than those of 50 controls with *SMN1*(+), without overlapping values. The mean ± SD Cq values of SMA patients and controls were 19.0 ± 1.4 and 10.1 ± 1.2, respectively. In the present study, we determined that Cq values ≥14 indicated the absence of *SMN1*.

Compared with the results of PCR–RFLP using DNA from freshly collected blood, the results from real-time mCOP-PCR using DBS for detection of *SMN1*-deletion showed a sensitivity of 1.00 (95% CI 0.96–1.00) and a specificity of 1.00 (95% CI 0.96–1.00) (Table 1).

### 3.2. Newborn Screening for SMA

Between January 2018 and April 2019, 4157 DBS samples were collected from newborn infants at 49 hospitals, covering 23 of the 47 prefectures in Japan (Figure 5). Of the 4157 collected samples, all tested negative for *SMN1* deletion using our new system described earlier. 

The quantity and quality of the collected DBS were not limiting factors for the detection of *SMN1* deletion in our system. Figure 6 shows two examples of good-quantity and -quality DBS (left) and poor-quantity and -quality DBS (right). Both clearly showed the presence of *SMN1* exon 7. Pre-amplification guaranteed that a sufficient amount of target sequence of *SMN* genes required for the identification of *SMN1* exon 7 deletion was generated. As a result, there were no samples with screening failure in the current study.

### 3.3. Follow-Up Study of the Infants Screened for SMA

Extraction survey results were obtained for 2370 babies from 17 hospitals. No infants showed developmental delay in motor milestones indicative of infantile-onset SMA at 6 or 10 months after DBS collection. 

## 4. Discussion

### 4.1. Targeted Pre-Amplification of SMN1/SMN2 Sequence

Our SMA screening system is an *SMN1*-deletion detection method that uses real-time mCOP-PCR technique following targeted pre-amplification of the sequence containing *SMN1/SMN2* from DBS DNA, which does not require any non-*SMN* reference genes for validation of PCR quality [18]. The pre-amplification product contains either *SMN1* or *SMN2*, or both. Here, *SMN1* and *SMN2* can be used as reference genes for each other, because all infants have at least one copy of *SMN1* or *SMN2* [9,10]. We used the pre-amplification product to confirm the presence of *SMN2* in the samples with the *SMN1*(–) genotype [18].

In the previous version of our system, genomic DNA was extracted from DBS before targeted pre-amplification [18]. However, the new version used in the present study did not include the DNA extraction step [20]. Instead, a punched circle from each DBS was placed directly into the reaction tube of the conventional PCR (direct PCR).

### 4.2. Modified Competitive Oligonucleotide Priming-PCR (mCOP-PCR)

In the mCOP-PCR, almost identical DNA sequences with one nucleotide difference (*SMN1* and *SMN2* exon 7 sequences in the present study) compete for annealing of the gene-specific oligonucleotide primer (SMN1-COP), and the better-matched DNA sequence (*SMN1* exon 7 sequence) is amplified much more efficiently. 

The original COP-PCR is a kind of allele-specific PCR in which two oligonucleotide primers with one nucleotide difference compete for annealing of the target DNA sequence in one PCR tube [24,25]. The core part of the original COP-PCR and our mCOP-PCR is the same: the lengths of the oligonucleotide primer sequences used in PCR amplification are shorter than the usual PCR primers and identical except for a single nucleotide difference that is located in the middle of the primer [24]. Thus, we used the term “modified COP-PCR” (“mCOP-PCR”) to refer to our gene-specific amplification method.

### 4.3. Accurate Detection System for SMN1 Deletion

In this study, firstly, we confirmed that the new version of our screening system has 100% specificity and sensitivity for the detection of *SMN1* deletion. The results were fully consistent with those of PCR–RFLP using DNA from freshly collected blood. Secondly, we successfully genotyped all DBS samples collected from 4157 newborn babies enrolled in the pilot study. There were no screening failures in the present study.

According to the real-time mCOP-PCR analysis in the validation study, the mean Cq values of SMA patients and controls were 19.0 ± 1.4 and 10.1 ± 1.2, respectively. This finding suggested that the amplification of the better-matched sequence (*SMN1* exon 7 sequence) was >100-fold more efficient than the amplification of the mismatched sequence (*SMN2* exon 7 sequence). Thus, we can claim that our SMA screening system using the real-time mCOP-PCR technique is an accurate method for the detection of *SMN1* deletion.

### 4.4. Robust System for SMA Newborn Screening Using DBS

To date, three other pilot studies have demonstrated the feasibility of population-based screenings of newborn infants for SMA in the USA [16], Taiwan [26], and Germany [27], and another study is underway in Belgium [28]. 

Of a total of 3826 infants screened in the US study and 120,267 infants in the Taiwan study, the first-pass assay failure rates of false-positive or false-negative were 3.0% and 0.04%, respectively [16,26]. The high first-pass assay failure rate in the US study was attributed to suboptimal DNA quality and quantity [16]. In the German study, in which 213,279 infants were screened, there were no false-positive or false-negative results; however, the authors described some invalid results in initial assessments, due to the DBS collection process or human errors (incorrect pipetting or incomplete sealing of the PCR plate) [27]. 

Of the 4157 newborn infants screened for SMA in our pilot study, there were no screening failures. Among the DBS collected from the hospitals in the present study, not all showed sufficient quantity and/or dryness of blood (Figure 6). However, we were able to determine the *SMN1* genotype of all newborns enrolled in the present study. The key to successful screening with our system may be targeted pre-amplification of *SMN* genes by direct PCR.

In our system, FTA^®^ Elute Cards could be replaced by other filter papers used for the screening of inborn errors of metabolism, such as phenylketonuria (PKU). We also confirmed that standard DBS filter paper other than FTA^®^ Elute Cards could work in our SMA screening system (data not shown).

### 4.5. Limitations of SMN1-Deletion Detection as an SMA Screening Strategy

Homozygous deletion of *SMN1* has been found in more than 95% of SMA patients, and intragenic mutations of *SMN1* have been found in the rest. Therefore, the primary purpose of SMA screening, at this moment, is to determine the presence or absence of *SMN1* [18]. None of the SMA screening studies, including ours, has detected any intragenic mutations in *SMN1*. For the detection of SMA patients with intragenic mutations, the screening system cannot be “simple, rapid, and inexpensive”; instead, a complex, time-consuming, and expensive system with next-generation sequencing may be necessary.

Our real-time mCOP-PCR system is specific for detecting *SMN1* deletions and does not provide any information about *SMN2* copy number. If required for prognostic purposes, *SMN2* copy number should be determined during a subsequent confirmatory assay or a second-tier assay [29,30]. However, *SMN2* copy number does not always correspond to disease severity, and factors other than *SMN2* copy number are also related to the severity of SMA [31].

## 5. Conclusions

Our *SMN1*-deletion detection system consists of a real-time mCOP-PCR technique following targeted pre-amplification of DNA from DBS, which rapidly and accurately detects *SMN1* exon 7 deletion even from DNA samples of poor quantity and quality. The current pilot study clearly demonstrated that our system is a useful method for newborn SMA screening, facilitating the early diagnosis of asymptomatic infants and allowing treatment to be started before irreversible motor neuron damage occurs. Thus, our newly developed system is ready to be applied in high-throughput SMA newborn screening.

## 6. Patents

*SMN1*-deletion detection system: Japanese patent application No. 2017-196967, PCT/JP2018/37732.

## Figures and Tables

**Figure 1 IJNS-05-00041-f001:**
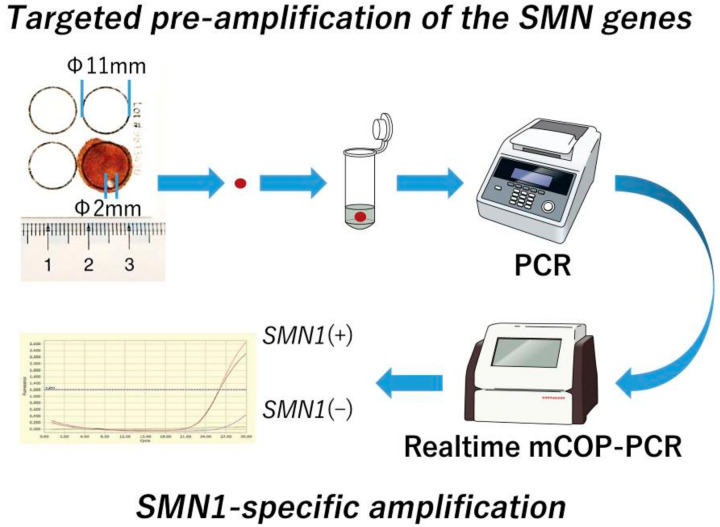
Scheme of the *SMN1*-deletion detection system using dried blood spots (DBS). Our system consists of two steps. Step (1) targeted pre-amplification of the *SMN* genes: the target sequences in the *SMN* genes from DBS are pre-amplified by conventional polymerase chain reaction (PCR), and Step (2) gene-specific amplification of *SMN1* and *SMN2* exon 7: *SMN1* deletion is detected by real-time modified competitive oligonucleotide priming-PCR (mCOP-PCR) with the pre-amplified products. In the first step, a punched circle from each DBS is placed directly into the reaction tube of conventional PCR (direct PCR).

**Figure 2 IJNS-05-00041-f002:**
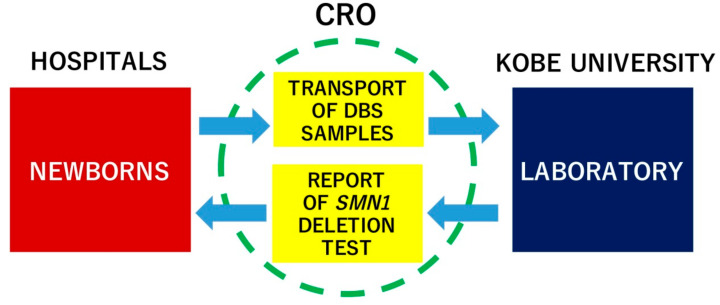
Data collection and data management flow between hospitals and Kobe University during the pilot study. CRO, contracted research organization.

**Figure 3 IJNS-05-00041-f003:**
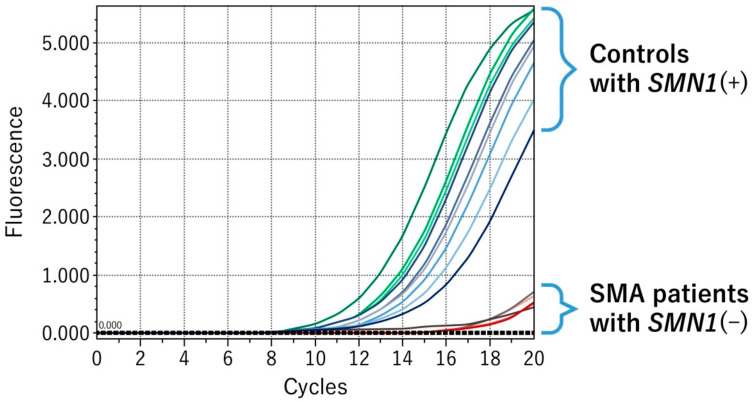
*SMN1*-specific amplification by real-time mCOP-PCR. SMA patients with *SMN1*(–) genotype showed no amplification with an *SMN1*-specific primer (SMN1-COP).

**Figure 4 IJNS-05-00041-f004:**
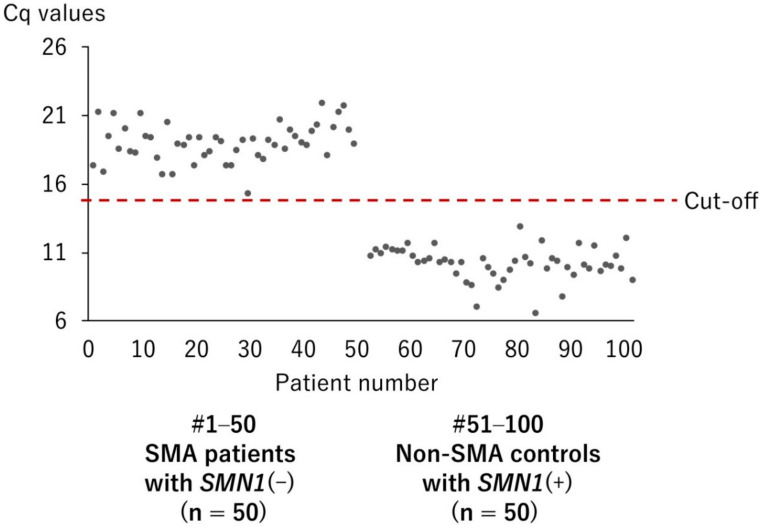
Distribution of quantification cycle number (Cq) values of DBS from 50 patients with *SMN1*(–) and 50 controls with *SMN1*(+). A Cq value of 14 was set as the cut-off point for the presence or absence of *SMN1*.

**Figure 5 IJNS-05-00041-f005:**
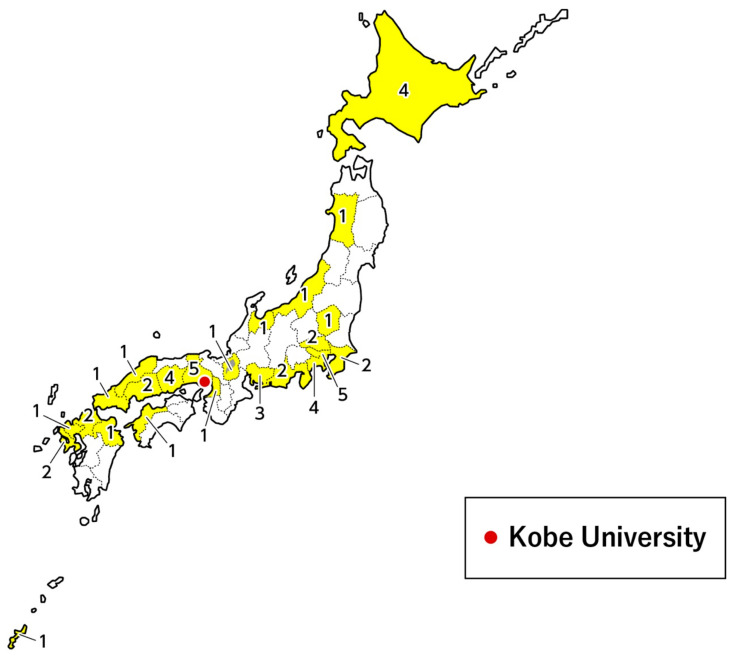
Location of the hospitals that participated in the pilot study. The numbers denote the number of hospitals in each prefecture that participated in the study.

**Figure 6 IJNS-05-00041-f006:**
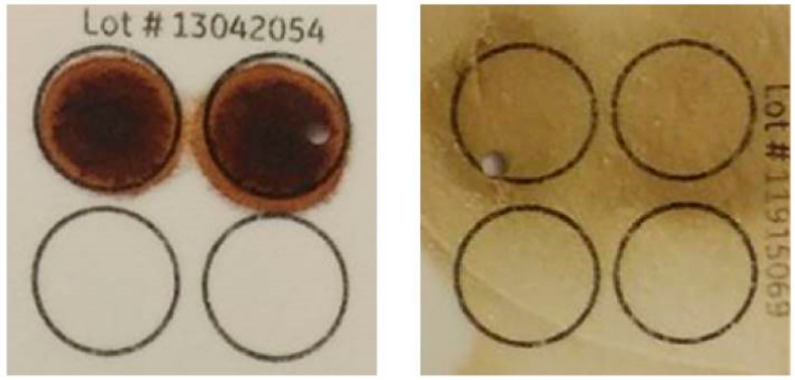
Two examples of good-quantity and -quality DBS (left) and poor-quantity and -quality DBS (right).

**Table 1 IJNS-05-00041-t001:** Real-time mCOP-PCR versus PCR–restriction fragment length polymorphism (PCR–RFLP) for the detection of *SMN1* deletion.

	PCR–RFLP (Fresh Blood)		Total
	*SMN1*(–)	*SMN1*(+)	
Real-time mCOP-PCR (DBS)			
*SMN1*(–)	50	0	50
*SMN1*(+)	0	50	50
Total	50	50	100

Sensitivity: 1.00 (95% CI 0.96–1.00); specificity: 1.00 (95% CI 0.96–1.00).

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
