# Peer review of "A Novel System for Spinal Muscular Atrophy Screening in Newborns: Japanese Pilot Study"

_2409-515X, 2019, doi:10.3390/ijns5040041_

Round 1

Reviewer 1 Report

The authors describe a real-time oligonucleotide PCR technique which can be used in dried blood samples to detect deletions of the SMN1 gene which may result in spinal muscular atrophy.

They demonstrate this using DBS from 50 SMA patients and 50 non-SMA controls, the results show acceptable sensitivity and specificity in this limited study.

They go on to apply this over a 15 month period analysing 4,157 DBS samples from newborns born in 49 hospitals.   There were no screening failures reported.   All samples tested negative and follow-up undertaken on 2,370/4,157 babies revealed no indications of SMA.

While this is a small study, the technique seems robust in practice and the retrospective study of 50 individuals with proven SMA demonstrated that all could be detected.

There have been a number of approaches described to detect deletions in the SMN1 gene that could be used in dried blood spots as part of a newborn screening programme so that is not novel but the paper is practical and topical as treatment for SMA is being accepted in many healthcare systems and early detection becomes a critical issue.

The paper is well written and illustrated.   I would recommend that it should be published in this form.

Reviewer 2 Report

[General Comment]

Recently, breakthrough treatment options for spinal muscular atrophy (SMA) like Nusinersen, have been developed. Timing to start such treatment is essential for the SMA children to achieve beneficial effects. Early detection will be essential, and neonatal screening will be significant. The authors developed a new breakthrough technology of screening for SMA, which uses the real time PCR after preamplification of DNA in blood spot disc of Guthrie card, and can easily be incorporated to the traditional NBS system using Guthrie card. The data is not influenced by extracted DNA quality, and both sensitivity and specificity of the data are almost perfect. The method will make a great contribution to establish neonatal mass screening for SMA for which treatments have been developed. This manuscript is worthy to be published in IJNS.

[Minor comments]

The author may give comments how much the test fee is estimated at present point, how many samples can be screening a day, or week points to be solved, if they have, in Discussion section.